# Uncovering the spin ordering in magic-angle graphene via edge state equilibration

Jesse C. Hoke [1,2,3], Yifan Li[1,2,3], Julian May-Mann[1,4], Kenji Watanabe [5], Takashi Taniguchi [6], Barry Bradlyn [4], Taylor L. Hughes[4] & Benjamin E. Feldman [1,2,3] ✉

The flat bands in magic-angle twisted bilayer graphene (MATBG) provide an especially rich arena to investigate interaction-driven ground states. While progress has been made in identifying the correlated insulators and their excitations at commensurate moiré filling factors, the spin-valley polarizations of the topological states that emerge at high magnetic field remain unknown. Here we introduce a technique based on twist-decoupled van der Waals layers that enables measurement of their electronic band structure and–by studying the backscattering between counter-propagating edge states–the determination of the relative spin polarization of their edge modes. We find that the symmetry-broken quantum Hall states that extend from the charge neutrality point in MATBG are spin unpolarized at even integer filling factors. The measurements also indicate that the correlated Chern insulator emerging from half filling of the flat valence band is spin unpolarized and suggest that its conduction band counterpart may be spin polarized.

The relative twist angle between adjacent van der Waals layers provides a powerful tuning knob to control electronic properties. In the limit of large interlayer twist, the misalignment leads to a mismatch in the momentum and/or internal quantum degrees of freedom of low-energy states in each layer, resulting in effectively decoupled electronic systems[1–7]. This decoupling can be sufficiently pronounced to realize independently tunable quantum Hall bilayers that support artificial quantum spin Hall states[2] or excitonic condensation[6,7]. In the opposite regime of low-twist angle, a moiré superlattice develops, and can lead to extremely flat electronic bands with prominent electron-electron interaction effects. The archetypal low-twist example is magic-angle twisted bilayer graphene (MATBG)[8–10], which has been shown to support symmetry-broken quantum Hall states[9,11–13] as well as correlated Chern insulators (ChIs) at high magnetic fields[11,14–22]. However, a full understanding of the nature of these states, including their spin and valley polarization, has so far remained elusive.

Combining large and small interlayer twists in a single device provides an approach to probe microscopic details of correlated ground states in moiré systems[23–25]. Such a device would yield electronically decoupled flat and dispersive bands, which can be used to interrogate each other. In some ways, this is reminiscent of other two-dimensional heterostructures which host bands of differing character. One notable example is mirror-symmetric magic-angle twisted trilayer graphene (MATTG) and its multilayer generalizations[26–32], which can be decomposed into flat MATBG-like bands that coexist with more dispersive bands. However, these bands hybridize at a nonzero displacement field, whereas a twist-decoupled architecture provides fully independent bands. This enables control over the relative filling of light and heavy carriers, including in a bipolar (electron-hole) regime. Crucially, in a perpendicular magnetic field, such a device can realize a quantum Hall bilayer with co- or counter-propagating edge modes. Because the inter-edge mode coupling depends on their respective internal degrees of freedom[2,33], the effects of edge backscattering on

[1]Department of Physics, Stanford University, Stanford, CA 94305, USA. [2]Geballe Laboratory for Advanced Materials, Stanford, CA 94305, USA. [3]Stanford Institute for Materials and Energy Sciences, SLAC National Accelerator Laboratory, Menlo Park, CA 94025, USA. [4]Department of Physics and Institute for Condensed Matter Theory, University of Illinois at Urbana-Champaign, Urbana, IL 61801, USA. [5]Research Center for Electronic and Optical Materials, National Institute for Materials Science, 1-1 Namiki, Tsukuba 305-0044, Japan. [6]Research Center for Materials Nanoarchitectonics, National Institute for Materials Science, 1-1 Namiki, Tsukuba 305-0044, Japan. ✉e-mail: bef@stanford.edu

transport can be used to identify spin/valley flavor polarization of the flat moiré bands.

Here we report transport measurements of a dual-gated, twisted trilayer graphene device that realizes electrically decoupled MATBG and monolayer graphene (MLG) subsystems. By tracking features in the resistance as a function of carrier density and displacement field, we demonstrate independently tunable flat and dispersive bands and show that transport measurements can be used to simultaneously determine the thermodynamic density of states in each subsystem. Furthermore, in the regime of counter-propagating MLG and MATBG edge modes in a magnetic field, we use longitudinal and non-local resistance measurements to infer the spin order within the MATBG subsystem–both for symmetry-broken quantum Hall states emanating from the charge neutrality point (CNP), and for the primary sequence of ChIs. Our work clarifies the microscopic ordering of correlated states in MATBG and demonstrates a powerful generic method to probe internal quantum degrees of freedom in two-dimensional electron systems.

## Results

### Twist-decoupled flat and dispersive bands

An optical image of the device is shown in Fig. 1a, with a side view of individual layers schematically illustrated in Fig. 1b. As we demonstrate below, the bottom two graphene layers have a twist of 1.11° and display behavior consistent with typical MATBG samples, while the topmost graphene layer is electrically decoupled because of the larger interlayer twist of ~5–6° (see Methods). The whole device is encapsulated in hexagonal boron nitride (hBN) and has graphite top and bottom gates. This dual-gated structure allows us to independently tune the total carrier density $n_{tot} = (C_b V_b + C_t V_t)/e$ and applied displacement field $D = (C_t V_t - C_b V_b)/(2\epsilon_0)$, where $C_{b(t)}$ and $V_{b(t)}$ are the capacitance and voltage of the bottom (top) gate, $e$ is the electron charge, and $\epsilon_0$ is the vacuum permittivity. The applied displacement field shifts the relative energies of states in each subsystem and, therefore, controls how the total carrier density is distributed between them (Fig. 1c).

We first describe electronic transport through the device at zero magnetic field. The longitudinal resistance $R_{xx}$ is largest along a curve at low/moderate $D$, with multiple fainter, S-shaped resistive features extending outward, i.e. approximately transverse to it (Fig. 1d). This phenomenology arises from electronic transport in parallel through the MLG and MATBG subsystems. Specifically, the strongly resistive behavior occurs when the MLG is at its CNP (solid black line in Fig. 1d). Relatively higher peaks in $R_{xx}$ along this curve reflect insulating states in MATBG. Analogously, when the carrier density in MATBG is fixed to an insulating state, $R_{xx}$ remains elevated even as the carrier density in the MLG is adjusted. This leads to the resistive S-shaped curves (such as the dashed white line in Fig. 1d; see discussion below).

The peaks in $R_{xx}$ centered near $n_{tot} = \pm 2.8 \times 10^{12}$ cm$^{-2}$ correspond to the single-particle superlattice gaps at moiré filling factor (number of electrons per unit cell) $s = \pm 4$. From these densities, we extract a twist angle of $\theta = 1.11°$ between the bottom two layers, and similar measurements using different contact pairs show that there is little twist angle disorder in these two layers (Supplementary Fig. 1). Intermediate resistance peaks are also present at $s = 0$, 1, $\pm 2$, and 3 (Fig. 1d, f). These peaks are consistent with the correlated insulators that have been previously observed in MATBG[8,12,13,34–39], and they persist as the MLG is doped away from its CNP (Supplementary Fig. 2). At higher temperatures, another peak develops near $s = -1$ (Supplementary Fig. 3), matching prior reports of a Pomeranchuk-like effect in MATBG[40,41].

Our characterization demonstrates the ability to independently tune the carrier density in each subsystem, and hence shows that the subsystems are effectively decoupled. This further allows the MLG to act as a thermodynamic sensor for the MATBG, similar to schemes in which a sensing graphene flake is isolated by a thin hBN spacer from the target sample[20,28,40,42]. By tracking the resistive maxima when the

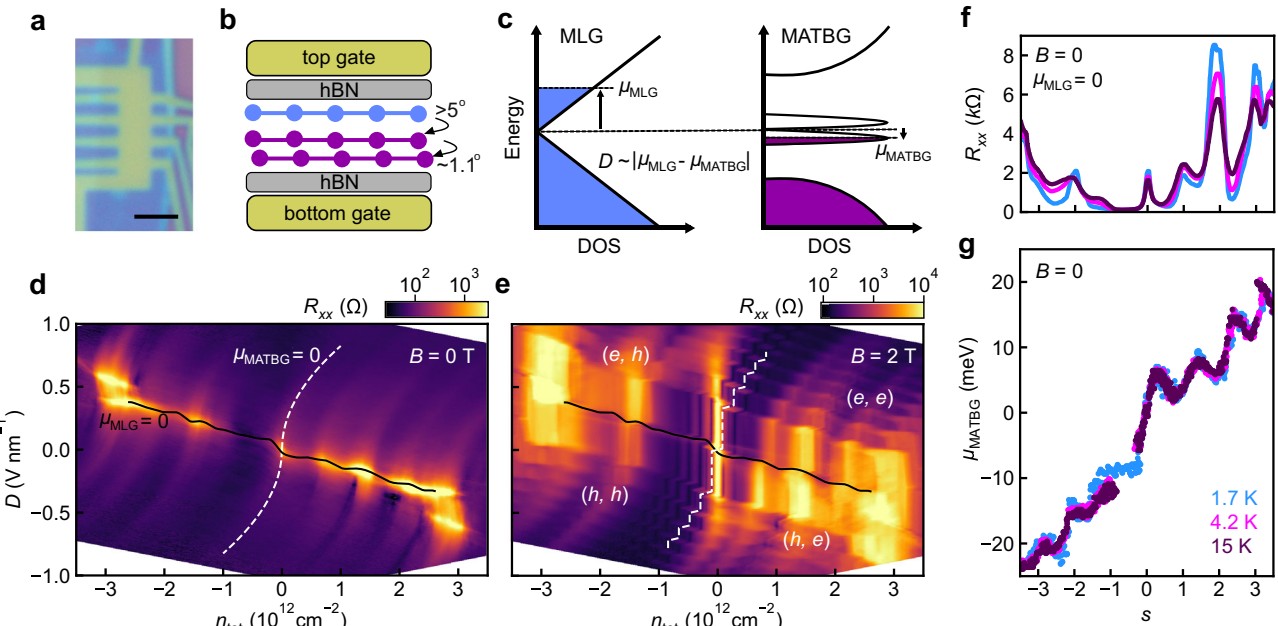

**Fig. 1 | Twist-decoupled monolayer graphene (MLG) and magic-angle twisted bilayer graphene (MATBG). a** Optical image of the device. The scale bar is 2 μm. **b** Schematic of the device structure and interlayer angles. The twisted trilayer graphene is encapsulated in hexagonal boron nitride (hBN) and has graphite top and bottom gates. **c** Band diagram of the combined MLG-MATBG system. The displacement field $D$ modifies the energies of states in each subsystem and, therefore, tunes the relative chemical potential $\mu_i$ of each subsystem $i$ at fixed total carrier density $n_{tot}$. **d, e** Longitudinal resistance $R_{xx}$ as a function of $n_{tot}$ and $D$, at zero magnetic field $B$ and at $B = 2$ T, respectively. Black solid (white dashed) lines denote where the MLG (MATBG) is at its charge neutrality point (CNP). Parentheses indicate which carrier types are present in the MLG and MATBG, respectively: $e$ indicates electrons and $h$ indicates holes. **f** $R_{xx}$ as a function of moiré filling factor $s$ at $B = 0$ and at various temperatures $T$ where the MLG is at its CNP (solid black curve in **d**). **g** $\mu_{MATBG}$ as a function of $s$ at $B = 0$, as extracted from (**d**) and analogous data at other temperatures.

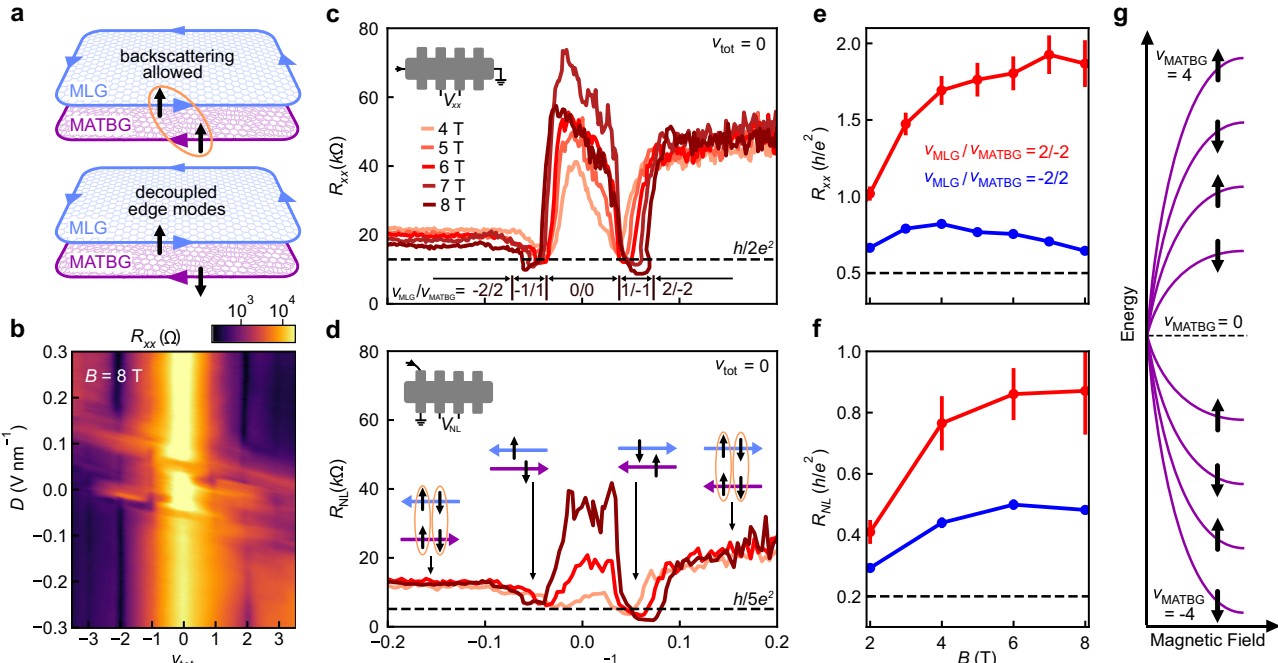

**Fig. 2 | Spin polarization of MATBG quantum Hall states near the CNP.**
**a** Schematic illustration of two possible scenarios for a single pair of counter-propagating edge modes. If the spins of each edge mode are aligned (top), back-scattering is allowed (orange circle). Backscattering is suppressed when the spins are anti-aligned (bottom), leading to quantum spin Hall-like behavior with $R_{xx} = h/2e^2$. **b** $R_{xx}$ as a function of the total filling factor $\nu_{tot} = \nu_{MLG} + \nu_{MATBG}$ and $D$ at $B = 8$ T. **c, d** $R_{xx}$ and $R_{NL}$, respectively measured in the configurations shown in the top left insets, as a function of $D$ when $\nu_{tot} = 0$. The filling factors of each subsystem for each regime of $D$ are indicated in the bottom inset of **c**. Insets in (**d**)

schematically represent the inferred relative spin orientations (black arrows) of edge modes in MLG (blue arrows) and MATBG (purple arrows), with orange circles indicating backscattering between a given pair. **e**, **f** $R_{xx}$ and $R_{NL}$ for $\nu_{MATBG} = \pm 2/\mp 2$ (red and blue, respectively) averaged over $0.1 < |D| < 0.25$ V nm$^{-1}$. Error bars correspond to one standard deviation. The straight lines connecting data points are guides for the eye. **g** Schematic diagram of CNP MATBG Landau levels (LLs) and their spin characters. Gaps between LLs are depicted schematically and do not represent experimentally measured field dependence.

MLG is at its CNP, and using a model that accounts for the screening of electric fields by each layer (Supplementary Note 2), we extract the MATBG chemical potential $\mu_{MATBG}$ (Fig. 1g). We find a total change of chemical potential across the flat bands of $\delta\mu \approx 40$ meV, with non-monotonic dependence on filling that matches previous reports of a sawtooth in inverse compressibility[14,21,40,41,43]. Similarly, we can determine the MLG chemical potential as a function of its carrier density $\mu_{MLG}(n_{MLG})$ by fitting it to the S-shaped resistive features in Fig. 1d, which occur at fixed $s$ in MATBG (Supplementary Note 2). These match the scaling $\mu_{MLG} \propto \text{sgn}(n_{MLG})|n_{MLG}|^{1/2}$ that is expected for the Dirac dispersion of graphene. We observe similar behavior in a second tri-layer device, where MLG-like states are decoupled from a bilayer gra-phene moiré system with a 1.3° twist angle (Supplementary Fig. 4), suggesting this is a generic phenomenon that is widely applicable in multilayer heterostructures.

Electronic decoupling is also evident when we apply a perpendicular magnetic field $B$, where the energy spectrum of MLG consists of Landau levels (LLs), and a Hofstadter butterfly spectrum develops in MATBG. Figure 1e shows $R_{xx}$ as a function of $n_{tot}$ and $D$ at $B = 2$ T, revealing staircase-like patterns which reflect crossings of the MLG LLs and MATBG states (Hall resistance $R_{xy}$ is plotted in Supplementary Fig. 5). Vertical features at constant $n_{tot}$ occur when the MLG is in a quantum Hall state; their extent (in $D$) is proportional to the size of the gap between LLs. As the displacement field tunes the relative energies of states in each subsystem, transitions occur when graphene LLs are populated or emptied. These cause each feature associated with a MATBG state to shift horizontally in density by the amount needed to fill a fourfold degenerate LL, $n_{LL} = 4eB/h$, where $h$ is Planck's constant and the factor of four accounts for the spin and valley degrees of freedom (e.g., see dashed white line in Fig. 1e).

## Quantum Hall edge state equilibration

In a magnetic field, the decoupled MLG and MATBG realize a quantum Hall bilayer in which either carrier type (electron or hole) can be sta-bilized in either subsystem. This results in co- (counter-)propagating edge modes when the respective carrier types are the same (different). Additionally, because the device is etched into a Hall bar after stacking, the edges of MLG and MATBG are perfectly aligned. Crucially, in the counter-propagating regime, the measured resistance encodes infor-mation about the efficiency of scattering between the edge modes in each subsystem (Supplementary Note 3), which depends on their internal quantum degrees of freedom. We expect that atomic scale roughness at the etched edge of the device enables large momentum transfer, and therefore anticipate efficient coupling irrespective of the valley (in MLG and MATBG) and moiré valley (in MATBG). However, assuming the absence of magnetic disorder, edge states having dif-ferent spins should remain decoupled, whereas those with the same spin can backscatter and exhibit increased longitudinal resistance (Fig. 2a). Probing $R_{xx}$, therefore, allows us to deduce the relative spin polarization of edge states in MLG and MATBG.

We first focus on low carrier density and high magnetic field, where the behavior of each subsystem $i$ is well described by quantum Hall states having filling factors $\nu_i = n_i h/eB$ emanating from their respective CNPs. A sharp peak in $R_{xx}$ emerges at combined filling factor $\nu_{tot} = 0$, flanked by several quantum Hall states at other integer $\nu_{tot}$ (Fig. 2b). These features exhibit a series of $D$-field tuned transitions as the relative filling of MLG and MATBG changes. The data encompass MLG states with $|\nu_{MLG}| \leq 2$. Importantly, prior work has shown that MLG edge modes at $\nu_{MLG} = \pm 1$ have opposite spin and valley quantum numbers, whereas those at $\nu_{MLG} = \pm 2$ are spin unpolarized[33]. Combin-ing this information with the measured resistance enables us to

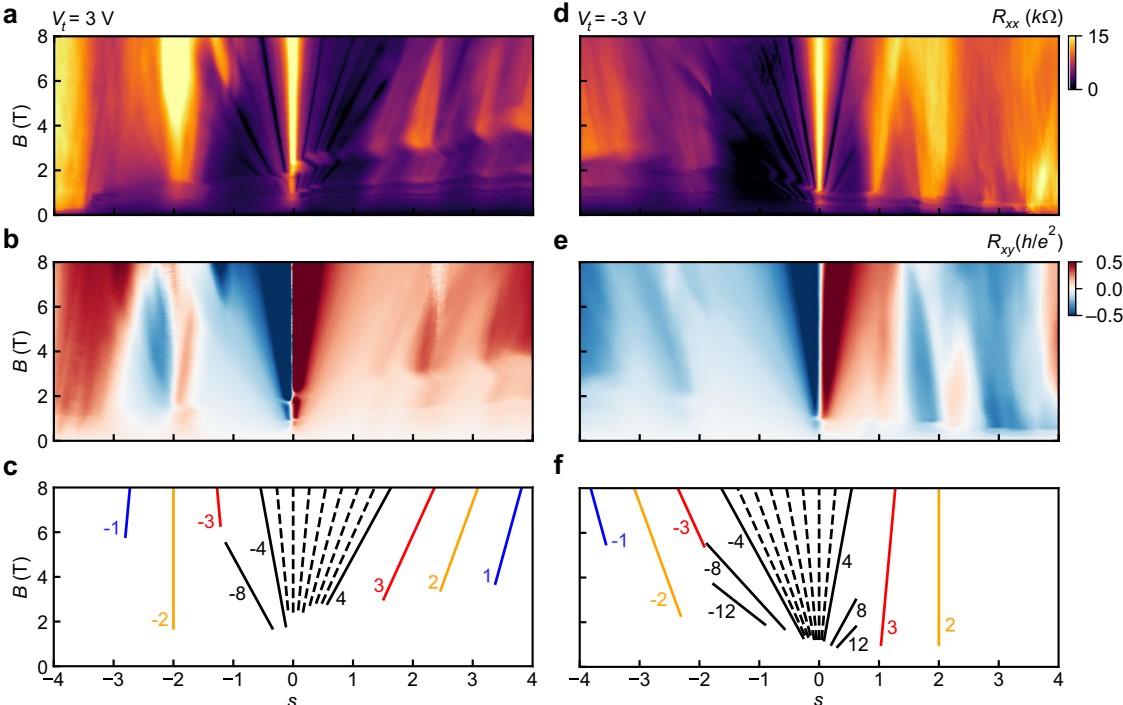

**Fig. 3 | Landau fans demonstrating correlated Chern Insulators (ChIs). a, b** $R_{xx}$ and $R_{xy}$ as a function of $s$ and $B$ at fixed top gate voltage $V_t = 3$ V. **c** Wannier diagram indicating the strongest quantum Hall and ChI states determined from (**a, b**). The Chern numbers $t$ of the MATBG states are labeled. At high fields, the total Chern numbers of each state are offset by 2 because $\nu_{MLG} = 2$. Black, red, orange, and blue lines correspond to states with zero-field intercepts $s = 0$, $s = |1|$, $s = |2|$, and $s = |3|$, respectively. For states with $s = 0$, $t \equiv \nu_{MATBG}$. Black dashed lines label the MATBG symmetry-broken quantum Hall states $-4 < \nu_{MATBG} < 4$. **d–f** Same as **a–c**, but for $V_t = -3$, where $\nu_{MLG} = -2$ at high fields. Data were collected at $T \approx 300$ mK.

determine the spin polarization of the MATBG quantum Hall states with $|\nu_{MATBG}| \leq 4$.

When $\nu_{tot} = 0$, MLG and MATBG have equal and opposite filling, and $R_{xx}$ approaches different values depending on the number of counter-propagating edge states (Fig. 2c). At $D = 0$, each subsystem is in an insulating, $\nu = 0$ symmetry-broken state. Here, no bulk conduction or edge modes are anticipated, and we observe a large resistance. Near $|D| \approx 0.05$ V/nm, $\nu_{MLG}/\nu_{MATBG} = \pm 1/\mp 1$, and $R_{xx}$ reaches a minimum near $h/2e^2$ (Fig. 2c). This phenomenology can be explained by a pair of counter-propagating edge modes with opposite spins, analogous to helical edge modes observed in large-angle twisted bilayer graphene[2]. This interpretation is further corroborated by similar behavior in another contact pair (Supplementary Note 4), and measurements of non-local resistance $R_{NL}$ (Fig. 2d). Indeed, the pronounced non-local resistance signal at $\nu_{MLG}/\nu_{MATBG} = \pm 1/\mp 1$ indicates that transport is dominated by edge modes (see Supplementary Note 5 for a discussion of bulk effects). This is corroborated by the value of $R_{NL}$, which is suppressed toward $h/5e^2$, the quantized value predicted from the Landauer–Büttiker formula for counter-propagating edge states in this contact configuration (Supplementary Note 3). We therefore conclude that similar to MLG, MATBG has a filled spin down (up) electron- (hole-)like LL at $\nu_{MATBG} = 1(-1)$.

Beyond $|D| \approx 0.08$ V/nm, where $\nu_{MLG}/\nu_{MATBG} = \pm 2/\mp 2$, we observe larger resistances $R_{xx} > h/2e^2$ and $R_{NL} > h/5e^2$ (Fig. 2c, d). This suggests that backscattering occurs for both pairs of edge modes: if both MATBG edge states had an identical spin, one counter-propagating pair would remain decoupled and would lead to quantized resistance $R_{xx} = h/2e^2$ and $R_{NL} = h/5e^2$ (Supplementary Note 3). A resistance above this value, as well as the large increase in resistance relative to $\nu_{MLG}/\nu_{MATBG} = \pm 1/\mp 1$, therefore both indicate that the edge states at $\nu_{MATBG} = \pm 2$ are spin unpolarized (see Supplementary Note 4, 5 for additional measurements and discussion of alternative interpretations which we rule out as unlikely). There is some asymmetry in the measured $R_{xx}$ depending on the sign of $D$; it is comparatively less

pronounced in $R_{NL}$. Since $R_{NL}$ is inherently a probe of edge conduction, this suggests the observed asymmetry in $R_{xx}$ originates from additional bulk current contributions, which may arise due to an electron-hole asymmetry in the strengths of different symmetry-broken states (see Supplementary Note 5). Based on the above observations, we deduce the spin polarization of the edge modes of the MATBG LLs emanating from its CNP, as illustrated in Fig. 2g.

**Addressing spin polarization of the Chern insulators**
In addition to symmetry-broken quantum Hall states emerging from the CNP, ChIs extrapolating to nonzero $s$ are evident in Landau fan measurements of $R_{xx}$ and $R_{xy}$ at fixed top gate voltages of $\pm 3$ V (Fig. 3). At these values, the MLG filling factor is $\nu_{MLG} = \pm 2$, respectively, at high fields. Consequently, both the Chern number of the primary sequence of quantum Hall states in MATBG (black lines in Fig. 3c, f) emerging from $s = 0$, and the ChIs (colored lines) are offset by $\pm 2$. After accounting for this shift, the ChIs that we observe are consistent with the primary sequence $|t + s| = 4$ commonly reported in MATBG, where $t$ is the Chern number of the MATBG subsystem[11,14–20]. Below, we focus primarily on the $(t, s) = (\pm 2, \pm 2)$ ChIs, which exhibit near-zero $R_{xx}$ and quantized $R_{xy}$ in the co-propagating regime (Supplementary Fig. 6). Here, ChI edge mode chirality is determined by the sign of $t$: states with $t > 0 (t < 0)$ have electron- (hole-)like edge modes.

Tuning into the bipolar (electron-hole) regime, allows us to realize counter-propagating edge modes from the MATBG ChIs and the MLG quantum Hall states. We apply the edge state equilibration analysis to determine the spin polarization of the ChIs in MATBG. For the $(t, s) = (-1, -3)$ ChI, we find a sharp resistive feature that occurs only when $\nu_{MLG} = 1$ (Fig. 4a, b), i.e., when there is one pair of counter-propagating edge states. The resistance grows with increasing $B$ and reaches values significantly larger than $h/2e^2$ (Fig. 4b). This indicates strong backscattering between edge modes, and hence that both have the

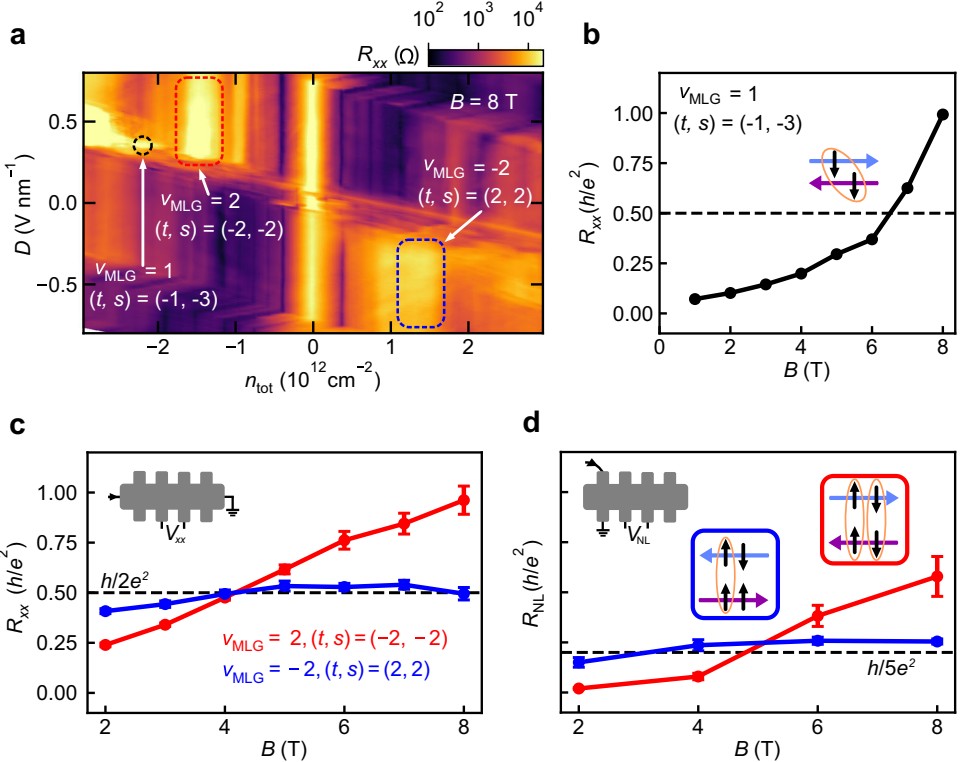

**Fig. 4 | Spin polarization of the ChIs in MATBG. a** $R_{xx}$ as a function of $n_{tot}$ and $D$ at $B = 8$ T (see Supplementary Fig. 8 for the equivalent map in a non-local contact configuration). Black dashed circle: $\nu_{MLG} = 1$, $(t, s) = (-1, -3)$. Red dashed box: $\nu_{MLG} = 2$, $(t, s) = (-2, -2)$. Blue dashed box: $\nu_{MLG} = -2$, $(t, s) = (2, 2)$. **b** $R_{xx}$ for the $\nu_{MLG} = 1$, $(t, s) = (-1, -3)$ state as a function of $B$. **c**, **d** $R_{xx}$ and $R_{NL}$, respectively, measured in the configurations shown in the top left insets, for $\nu_{MLG} = \pm 2$, $(t, s) = (\mp 2,$ $\mp 2)$ states (red and blue, respectively) as a function of $B$. Data are averaged over $0.325 < |D| < 0.525$ V nm$^{-1}$. Error bars correspond to one standard deviation. Insets in (**d**) schematically represent the inferred relative spin orientations (black arrows) of edge modes in MLG (blue arrows) and MATBG (purple arrows), with orange circles indicating backscattering between a given pair. The straight lines connecting data points are guides for the eye.

same spin (inset, Fig. 4b). We conclude that the first flavor to occupy the MATBG Hofstadter subbands (see Supplementary Note 6) is spin down, consistent with expectations based on the Zeeman effect.

A resistive state also occurs when $(t, s) = (-2, -2)$ and $\nu_{MLG} = 2$ (Fig. 4a). We observe $R_{xx} > h/2e^2$ that grows with increasing $B$ (Fig. 4c and Supplementary Fig. 7), indicating efficient backscattering between both pairs of counter-propagating edge modes. We obtain consistent results from both the non-local resistance (Fig. 4d) and $R_{xx}$ measurements of a second contact pair (Supplementary Note 4). We, therefore, conclude that the $(-2, -2)$ ChI in MATBG is spin unpolarized (red inset, Fig. 4d).

In contrast, we observe more moderate resistance for the $(t, s) = (2, 2)$ ChI in MATBG when $\nu_{MLG} = -2$ (Fig. 4a). In measurements of $R_{xx}$ ($R_{NL}$) at fixed $B$, the resistance of this state saturates near $h/2e^2$ ($h/5e^2$) at high $B$ (Fig. 4c, d), with similar near-quantized $R_{xx}$ in a Landau fan measurement (Supplementary Fig. 7). Together, these results demonstrate that there is only partial coupling between edge modes. The data are consistent with one pair of decoupled, counter-propagating edge modes, and another pair having allowed backscattering. This would naturally arise if the $(t, s) = (2, 2)$ ChI in MATBG is spin polarized (blue inset, Fig. 4d). The data, therefore, suggest a spin polarized ground state may be favored (see Supplementary Notes 5, 6 for further discussion).

## Discussion
The observed spin orderings of both the quantum Hall states and the ChIs clarify the microscopic interactions and relative strengths of different symmetry breaking terms in MATBG. Near charge neutrality, spin unpolarized states are favored at $\nu_{MATBG} = \pm 2$ for all measured magnetic fields $B > 2$ T (Fig. 2e, f). This is counter to expectations based on both the conventional Hofstadter subband model (see

Supplementary Note 6) and Zeeman considerations. Specifically, moiré valley splitting[14], which arises in the presence of $M_y$ symmetry breaking, or some other mixing between Hofstadter subbands is necessary to produce a spin unpolarized state at $\nu_{MATBG} = \pm 2$ (see Supplementary Note 6). Moreover, even in the presence of moiré valley splitting, the Zeeman effect would favor spin polarization at $\nu_{MATBG} = \pm 2$; our observations therefore indicate that exchange interactions dominate over Zeeman splitting throughout the measured field range and favor spin unpolarized states.

Very recent theoretical work[44] suggests there is a crossover between spin polarized ChIs favored by the Zeeman effect at high magnetic field and a partially spin unpolarized intervalley coherent state favored at low magnetic field, with the former predicted to dominate at experimentally relevant fields. Our results for the $(t, s) = (2, 2)$ ChI are consistent with the high field prediction, but the spin unpolarized states we observe at $(t, s) = (-2, -2)$ are not. This discrepancy likely reflects electron-hole asymmetry in MATBG and/or atomic scale relaxation of the lattice, which are neglected in the theoretical model. The calculations indicate close competition between different ground states, so including these effects will alter quantitative predictions and could even lead to qualitatively different ground states, as observed experimentally. Our work provides an important benchmark for future theoretical considerations, demonstrating the importance of these terms, that distinct spin ordering can occur for electron and hole doping, and that antiferromagnetic exchange contributions can be comparable to or larger in magnitude than Zeeman splitting.

In conclusion, we have realized a twisted graphene multilayer consisting of electrically decoupled MATBG and MLG subsystems. Even though the layers are in contact, we demonstrated that a twist-decoupled architecture provides a method to extract thermodynamic

properties and probe internal quantum degrees of freedom through edge state equilibration. Looking forward, we anticipate its extension to other van der Waals materials, including to recently discovered systems that exhibit fractional quantum anomalous Hall states[45–49]. This device geometry also represents the most extreme limit of dielectric screening of interactions[34–36] in which a tunable screening layer is immediately adjacent to the system of interest. More generally, it provides a natural arena to explore Kondo lattices[50,51] with independently tunable densities of itinerant electrons and local moments, as well as an opportunity to study Coulomb drag between adjacent layers[52].

## Methods

### Device fabrication
The MATBG-MLG stack was fabricated using standard dry transfer techniques with poly (bisphenol A carbonate)/polydimethylsiloxane (PC/PDMS) transfer slides[53,54]. A monolayer graphene flake was cut into three pieces with a conductive AFM tip in contact mode. An exfoliated hBN flake (26.5 nm) was used to sequentially pick up each section at the desired twist angle before placing it on top of a prefabricated stack of few-layer graphite and hBN (27 nm). Finally, an additional few-layer graphite flake was added and served as the top gate. The stack was subsequently patterned with standard electron beam lithography techniques followed by etching to form a Hall bar geometry and metallization to form edge contacts[53].

### Transport measurements
Transport measurements were conducted at cryogenic temperatures (1.7 K unless otherwise stated) using standard lock-in techniques with a current bias of 5–20 nA at 17.777 Hz. Because edge contacts are made to the etched sample, they simultaneously make electrical contact to all three graphene layers, and electronic transport through the device reflects parallel transport through both MATBG and MLG subsystems. The longitudinal and transverse resistance are symmetrized and anti-symmetrized in a magnetic field, respectively.

### Determination of twist angle
The twist angle $\theta$ between the pair of layers that form MATBG is determined by the superlattice carrier density $n_s = 4/A \approx 8\theta^2/\sqrt{3}a^2$ when the MLG is at charge neutrality. Here, $A$ is the superlattice unit cell area and $a = 0.246$ nm is the MLG lattice constant. The twist angle can also be independently confirmed by fitting Chern insulators in a Landau fan measurement using the Streda formula $dn/dB = C_{\text{tot}}/\Phi_0$, which have intercepts at zero magnetic field at integers $s = 4n/n_s$. Both methods yield a consistent value of $\theta = 1.11^o \pm 0.05^o$, where the quoted uncertainty reflects the width of the $s = \pm4$ resistive peaks. The capacitances $C_b$ ($C_t$) between the bottom (top) gate and sample are accurately determined based on the slopes of features in Landau fans taken at constant bottom (top) gate voltages $V_b$ ($V_t$). This also results in vertical features (Fig. 1e, for example) when data are plotted as a function of total carrier density $n_{\text{tot}} = C_t V_t/e + C_b V_b/e$ and displacement field $D = (C_t V_t/e - C_b V_b/e)/(2\epsilon_0)$, where $e$ is the electron charge, and $\epsilon_0$ is the vacuum permittivity.

The relative angle between the MLG and MATBG subsystems is estimated based on the angle between the AFM-cut edges of the top and middle graphene layers. An optical image of the three graphene (and top hBN) layers on the PC/PDMS stamp during the stacking process (before deposition) is shown in Supplementary Fig. 9. From the image, we identify a twist angle between the top and middle graphene layers of about 5.5° ± 0.5°.

## Data availability
Data that support the findings in this study are available at https://doi.org/10.5281/zenodo.11044381. Additional datasets generated and/or analyzed during the current study are available from the corresponding author upon request.

## Code availability
The codes that support the findings of this study are available from the corresponding authors upon request.

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

## Acknowledgements
We thank Pablo Jarillo-Herrero, Steve Kivelson, Yves Kwan, Sid Parameswaran, B. Andrei Bernevig, Oskar Vafek, Xiaoyu Wang, David Goldhaber-Gordon, and Aaron Sharpe for helpful discussions. This work was supported by the QSQM, an Energy Frontier Research Center funded by the U.S. Department of Energy (DOE), Office of Science, Basic Energy Sciences (BES), under Award # DE-SC0021238. K.W. and T.T. acknowledge support from the JSPS KAKENHI (Grant Numbers 20H00354 and 23H02052) and World Premier International Research Center Initiative (WPI), MEXT, Japan. J.C.H. acknowledges support from the Stanford Q-FARM Quantum Science and Engineering Fellowship. Part of this work was performed at the Stanford Nano Shared Facilities (SNSF), supported by the National Science Foundation under award ECCS-2026822.

## Author contributions
J.C.H. fabricated the devices. J.C.H. and Y.L. conducted transport measurements. B.E.F supervised the project. J.M.M., B.B., and T.L.H. provided theoretical support. K.W. and T.T provided hBN crystals. All authors contributed to analysis and writing of the manuscript.

## Competing interests
The authors declare no competing interest.
