## [Peer Review File · Nature Communications]

Uncovering the spin ordering in magic-angle graphene via edge state equilibrationEditorial Note: This manuscript has been previously reviewed at another journal that is not operating a transparent peer review scheme. This document only contains reviewer comments and rebuttal letters for versions considered at Nature Communications.

Reviewers' Comments:

Reviewer #1:

Remarks to the Author:

All of my concerns have been addressed and I recommend publication in Nature Communications.

Reviewer #2:

Remarks to the Author:

I thank the authors for their modifications to the paper based on my prior comments and for the discussion in their detailed response. I believe that the refined paper, particularly with the addition of substantial text to the "Discussion" section nicely describes the impact of the observations of spin polarization at filling factor zero and at $(t,s)=(-2,-2)$ and spin polarization at $(t,s)=(2,2)$. I am impressed with both the new technique developed here and in the thoroughness of the paper. I also appreciate the additions to the figures that now make the paper easier to understand. I have no further issues with the paper, and I believe that it should be published without delay in Nature Communications.

REVIEWERS' COMMENTS

Reviewer #1 (Remarks to the Author):

All of my concerns have been addressed and I recommend publication in Nature Communications.

We thank the reviewer for their constructive comments and their recommendation for publication in Nature Communications.

Reviewer #2 (Remarks to the Author):

I thank the authors for their modifications to the paper based on my prior comments and for the discussion in their detailed response. I believe that the refined paper, particularly with the addition of substantial text to the "Discussion" section nicely describes the impact of the observations of spin polarization at filling factor zero and at $(t,s)=(-2,-2)$ and spin polarization at $(t,s)=(2,2)$. I am impressed with both the new technique developed here and in the thoroughness of the paper. I also appreciate the additions to the figures that now make the paper easier to understand. I have no further issues with the paper, and I believe that it should be published without delay in Nature Communications.

We thank the reviewer for their constructive comments and their recommendation for publication in Nature Communications.